# Conceptual and Preliminary Design of a Shoe Manufacturing Plant

**Jorge Borrell Méndez** [1,*] **, David Cremades** [2] **, Fernando Nicolas** [2] **, Carlos Perez-Vidal** [1] **and Jose Vicente Segura-Heras** [3]

1   Systems Engineering and Automation Department, Miguel Hernandez University, 3202 Elche, Spain; calos.perez@umh.es
2   Simplicity Works Europe, 3202 Elche, Spain; d.cremades@simplicityworks.es (D.C.); fernando.nicolas@simplicityworks.es (F.N.)
3   Operations Research Center, Miguel Hernandez University, 3202 Elche, Spain; jvsh@umh.es
*   Correspondence: jorge.borrell@goumh.umh.es

**Abstract:** This article presents a procedure for designing footwear production plants with a Decision Support System combined with an expert system and a simulation approach. The footwear industry has many operations and is labour intensive. Optimisation of plant layout, machinery, and human resources is very important to design the footwear manufacturing system, making adequate investment in space and equipment. In the industry it is essential to reduce the process time, so the research is based on a Decision Support System combined with an expert system and simulation to improve the design of the manufacturing plan. This work contains two case studies, direct injection manufacturing and assembly and carburising methods, which are compared to analyse all the necessary resources to have the best cost–benefit ratio. In each case, a precise knowledge of the type and quantity of machinery and human resources is needed to estimate the production. This comparison has been done through simulations and using a knowledge base of an expert system. The conclusions are presented in which an improvement in production time is obtained by applying the methodology developed in the study.

**Keywords:** expert system; manufacturing system design; shoe industry; direct-injection; simulation

## 1. Introduction

The design of a manufacturing plant is an important feature, more so at present in which automation is an upward trend. Nevertheless, the correct design of the plant is essential both in a low automation plant and in a plant with a high degree of automation. The position of the machines and the different process steps can reduce the time of the whole process, and this is studied little in the literature for the specific case of the shoe industry. It is essential that the process follows a fluent line in all steps. Nowadays, in most of the manufacturing plants and in particular in the shoe industry, the design of the manufacturing plant is based on the experience of the worker. The shoe industries are a low automatised industry compared with other sectors and the increase of automation in recent years could make the worker's experience in designing the plant incomplete, due to the processes that become fully automated. For this main reason, the shoe industry presents high difference with other industries, a reconversion of the industry is being carried out to automatise it, but this process is gradual, applying known techniques to a non-exploited industrial sector with the aim of full automatisation.

For this reason, this research presents a more scientific method to design a manufacturing plant combining different elements such as Decision Support System combined with expert systems and simulations. It seems coherent to tackle this problem using an expert system approach. Using a predetermined set of rules based on the authors experience, the core of a rule-based system can be described. These rules are mainly obtained from technical bibliography and expert knowledge. Due to the high degree of handicraft in this sector,

the expert system is based on a knowledge base previously collected from expert workers in the footwear sector. The use of this knowledge is essential as there is no precedent in highly automated footwear manufacturing plants. This is the main reason for the use of a Decision Support System combined with expert system in the development of this study. There are several features such as flow description, relationship and dependencies between processes, task times (including statistical distribution), or utilisation rate that are required to build the Expert System. All of them are presented by A. Kidd in [1]. In this project, the knowledge acquisition process was mainly provided by co-authors of this work and experts in the footwear industry. The knowledge provided by the experts was completed with bibliographic information. However, due to the complexity of the production process, a set of rules are not enough to define all relationships between processes and resources and part of the knowledge is provided through simulations of the plant. This work presents a decision support system which is combined with an expert system, using the knowledge of an experienced group of people to improve the final shoe plant design, supporting the design decisions and where human and material resources and real processing times have been considered.

In this work, two case studies were considered. The case of direct-injection and mounting-and-cementing processes were studied and they encompass around 90% of the world shoe production. The expert system is quite similar in both cases, but specific differences were considered in their respective implementation on the simulator.

This expert system can be tested using discrete element simulation to verify the performance and results provided by the implementation. Computer simulation techniques are more and more used in the industry to reduce production costs by optimising resources. These simulation techniques in most cases require a complex mathematical model of the production plant but alternative models can be used in order to adjust a trial-and-error approach and to determine which configuration suits the purpose best. The simulation used in production systems may have different levels of complexity. The task can be carried out without a visualisation package or through a 3D animated modelling. Regarding the accuracy of the model, it can contain a simple mathematical model developed in a few hours or an enormously detailed and complex model that requires years of work. The precision level of the model could be decided in purely economic terms, the savings as a result of the simulation should be greater than the cost of carrying out the simulation [2]. However, it is not always possible to quantify the savings directly, especially when the savings cannot be quantified until the simulation has been performed. Discrete events simulation is widely applied in components manufacture, subsets, and final product in different industries. There are many different packages available for simulating discrete events, each with different characteristics and a different cost price. The potential user has to select among many available packages the one that best suits the requirements of his application. The selection of simulation software needs a clear definition of features that allow to make a decision on what software should be used. All simulation programs can be used to examine material costs, process programming, product assembly strategies, warehouse dimensioning, layout distribution, logistics strategies, etc. Such tools can provide useful information, which could help to evaluate production and logistics helping to optimise the design of the production process. In this project, the plant distribution, machinery, and human resources analysis were carried out using Anylogic, see [3–5] for more information.

The paper is structured as follows: Section 2 presents specific features of the shoe manufacturing industry and process. Section 3 presents a literature review meanwhile Section 4 presents the methodology and materials including the expert system and the software used to design the factory. Sections 5 and 6 describe two case studies that represent around the 90% of the shoe production worldwide. Finally, Section 7 describes a comparative between both processes studied, and Section 8 shows the conclusions of this work.

## 2. The Shoe Industry

In general, the global footwear market has enjoyed a healthy and stable growth rate in recent years, and it is expected to continue this trend soon. The global footwear market is expected to expand at an overall annual compound growth rate (CAGR) of 3.44% from 2018 to 2023, leading to a global revenue of USD 280.61 billion by 2023. The overall production of shoes is close to 24.3 billion pairs of shoes [6]. Expected growth in the US is around 1.8%. Within Europe, which continues to remain an important element in the world economy, Germany will add over USD 2.4 billion to the region's size and clout in the next 5–6 years. Over USD 1.8 billion worth of projected demand in the region will come from of the remaining Europe markets. Casual shoes, one of the segments analysed in the paper of Reddy [6], will reach a market size of USD 11.1 billion only in Japan by the close of 2023. On the other hand, China exhibits the potential to grow at 3.5% over the next couple of years. Several macroeconomic factors and internal market forces will shape growth and development of demand patterns in emerging countries in Asia-Pacific, Latin America, and the Middle East.

*Shoe Production*

Shoe manufacturing has not suffered significant changes over the years. Despite the recent use of machinery, the process is mainly handmade, very laborious (around 80 different operations), and especially complex, thus making labour work cost one of the most important parts of footwear cost structure [7]. Figure 1 shows some of the tasks commonly performed in the shoe industry. The main steps performed are as follows:

- Cutting: The different pieces (of leather, synthetic canvas, suede, textile, etc.) that will form the upper are cut (Figure 1(A5)). In addition, cut pieces are prepared for the following processes by doing guiding marks or temporary unions among others.
- Stitching: Upper pieces are stitched together with a sewing machine (Figure 1(B5–A4)).
- Assembly: Upper is prepared by placing heel counter, welt, and toecap if required. Some pieces are pre-shaped and mounted using a last and then, the upper is bonded to the outsole (Figure 1(B4–C1)).
- Outsole attachment (or injection): The outsole is attached to the upper (mounting-and-cementing method) or it is injected directly to the upper (direct injection method). Figure 1 shows the mounting-and-cementing method in Figure 1(F3, A1, B1).
- Finishing: In this final step, the product gets its final appearance. Processes of cleaning, insole, and laces placement, etc., are done. Finally, the product is packed and stored in the warehouse (Figure 1(D1–I1)).

The most common shoe production method is called mounting-and-cementing. In this case, the shoe outsole is glued to the upper using a special machine to reactivate the glue (cement) and to apply a high pressure during few seconds, this is a traditional and former process as described in [8]. The alternative method that is currently gaining ground because of efficiency and product quality is called direct-injection process [9,10]. In this method, the sole is directly injected over the upper creating a strong union between materials, thus making footwear more durable. Cutting and stitching operations in the upper are done exactly in the same way as for the mounting-and-cementing process but when the upper is prepared, it is placed in an injection machine that will create the outsole permanently bonded to the upper. Usually, direct-injection process uses a rotary machine (see Figure 2) with multiple mould stations in order to allow the placement of the lasted upper. One or two injections can be done, and the curing process is performed according to the machine movement. In both processes, robots are used for sanding of the base leather in order to increase the bonding of the sole, to apply glue for bonding, or to pick up the pieces to the mould [11–13].

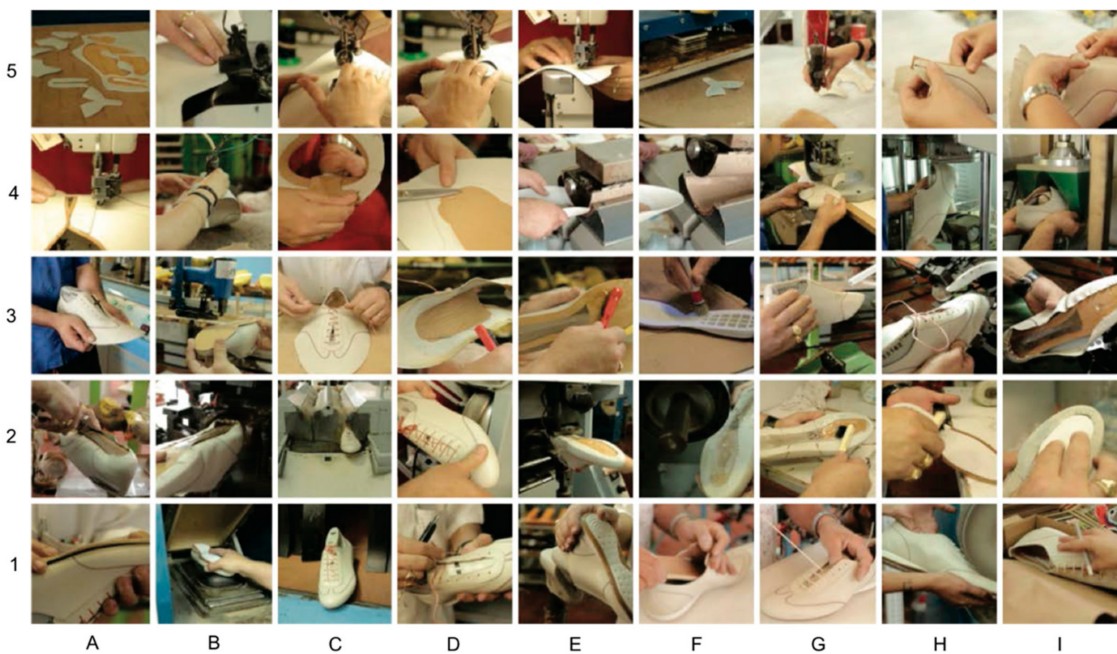

**Figure 1.** Steps to produce a pair of shoes. Courtesy of Simplicity Works Europe.

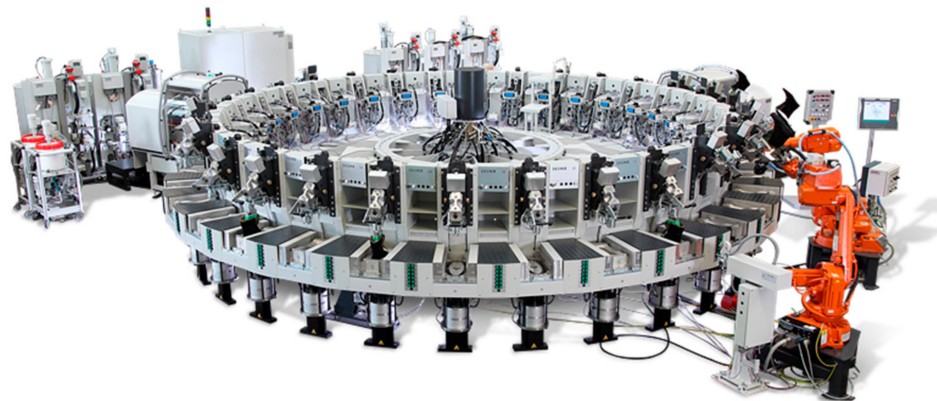

**Figure 2.** DESMA rotary machine equipped with two injectors and robotic cells. Courtesy of DESMA Schuh-maschinen GmbH.

In each case, the key element of the production process is the outsole attachment or injection. This step of the production is what gives the name to the whole manufacturing process (direct injection or mounting-and-cementing) and they are the factory bottlenecks or limiting factors that should be optimised. The last operation in the manufacturing process is the attachment of the outsole, together with its previous operation, the gluing of the upper and the outsole itself. The outsole and the upper can be glued manually by using a brush impregnated with adhesive or by using a machine or a robot, but in any case, assuring that the application of the adhesive is uniform over the entire surface. Once the outsole and the upper have been glued, the adhesive must be reactivated. This reactivation process is carried out in a cabin, provided with a conveyor belt, where heat is generated for a certain time to the area of the outsole and the upper where the glue has been applied. When the adhesive of the outsole and the upper has been reactivated, it is time to attach the outsole to the upper with precision so that both are perfectly centred. Next, a press-type machine with a deformable base is used. The pressure and the angle of the deformable elements are regulated according to the type and shape of the shoe. The machine works by hydraulic action, pressing the upper towards a support that descends and holds the outsole, generating a great pressure between the two elements, which remain stuck after a few seconds.

The economic cost of the rotary injection machine is significantly higher than the equivalent machinery to perform the mounting and cementing process, but it has two big advantages. On the one hand, the productivity of direct injection is greater than the mounting and cementing process. On the other hand, the quality of the product to be marketed justifies an increment in the product price, which facilitates the return of the investment performed on machinery. In relation to the labour cost, salaries in some Eastern European countries and even in some areas of Portugal are already competitive compared to the average labour cost in China (although not at the moment in Vietnam). This makes the production of footwear in Europe a more and more attractive option for production companies.

There is some research focused on the plant design, in [14] the design is focused on the workers ergonomics and in the paper of Ulutas [15] the design is focus on the dynamic distribution in shoe manufacturing plants. Therefore, to optimise the layout and resources of a shoe production factory, expert knowledge and queue modelling are very good options because it can help to adjust this labour-intensive process where the investment on machinery makes some decisions critical. In other words, by using expert systems and a discrete element simulation tool, the whole manufacturing process can be defined to get the best cost–benefit ratio.

## 3. Literature Review

There is some research about the expert systems in the industry. It is an important tool in order to improve, more efficiently, the processes. In the paper of Linko [16], that advantages that an expert system could have in the food industry is explained. In recent years, with the growth of artificial intelligence, expert systems are considered an approach to artificial intelligence and are used to improve their solutions in [17–19]. While [17] is focused on a review of the chemical industry, [18] is focused on detection of food adulteration. In the case of [19], Jha et al. present also a review, but in this case focused on automation in agriculture. Therefore, expert systems are widely used in the industry to automatise and improve the process. The shoe industry is highly competitive due to globalisation in production and the number of big companies competing in the same market. In this context, shoemakers have reached a level of competitiveness in which every detail is deeply analysed in order to get the best result or the highest productivity. Many times, the attention is focused on the layout design, machinery, and human resources. To analyse these factors, several techniques can be used and the most representative are described below. A relevant case is shown in [20], where the aim is to determine the optimum production policy over the combinations of the models which will be manufactured in daily working schedule when producing men's shoes. A simulation study developed using Arena to see at what degree the variations of the models affect the throughput rate is carried out and presented in that work. In [21], the main objective is to use a simulation tool to validate a problem of production line balancing, in a specific case that refers to the footwear industry, where the production will need to respond to customised requests by the customers regarding the footwear industry. The document presents three options to simulate the shoe production: Arena, Flexsim, and Simio. Simio is finally chosen to implement the simulations and perform the analysis. In [22], it is shown how to select the quantity of units to schedule for a shift duration, it computes the number of operators needed on a line, it sets the conveyor speed, and coordinates the main line with sub-assembly lines. It also assigns the work elements to the operators on the line and the sequence of the models down the line. It is important for the plant management to have a systematic way to determine the amount of lasts per model. In the paper of Bangsow [23], it is aimed at researchers who deal in their work with Discrete Event Simulation. In the paper of Nisanci [24], a simulation study of a shoe manufacturing plant consisting of seven departments is presented. This study is part of an investigation which was directed at understanding the characteristics of the production system and to assess its behaviour under alternative operating policies. The effectiveness of the plant is evaluated to select a performance

criterium. Other solutions like [25] are based on an Adaptive Large Neighbourhood Search (ALNS) heuristic simulation model to search for the problem domain. In that case, the authors present a case study of an industrial footwear manufacturing plant to demonstrate the effectiveness of the proposed approach. That is an alternative way to design the plant layout without using queue theory. The work presented by [26] proposes a scheduling module focused on the short term in order to respond quickly to market needs and changes in a flexible manner. The module is composed of a finite capacity scheduler integrated with a new software based on the Analytical Hierarchy Process (AHP) decision support system. This module considers the aspects related to order importance as complexity or urgency and assigns each order a priority.

## 4. Methodology and Materials

This research is based on an expert system design supported by discrete event simulation. Firstly, the expert knowledge collected is essential in order to create a database. This database will be used by the expert system to develop the design required. In this research, a shoe manufacturing plant is designed using both tools. Secondly, a simulation of the production plant with it will be used to verify the solution. On the other hand, a large amount of accurate information about the solution (task times, number of workers, etc.) will be gathered in the simulator.

### 4.1. Expert System Approach

A predetermined set of rules established by experts and based on the production process is the knowledge base used in this project. These rules are obtained from bibliographic sources and expert's knowledge. There are several techniques to establish rules as done in [27]. In this application, the knowledge was provided by coauthors of this work and by industrial process modelling tools.

External experts also participated providing valuable information about the production process, its characteristics, and its restrictions. These process modelling tools are necessary because not all knowledge can be provided by experts as a set of rules [28]. Some of the information necessary to represent the system is introduced into a simulation tool to allow the proper definition of entities, tasks, variables, and the relationship between them. In addition, design data needs to be organised in an ordered manner so that it is easily accessible and understandable to the user. The simulation model allows to represent the experts' knowledge in a very similar way to the physical software of the footwear production plant [28]. In this research, the inputs considered were mainly the expert knowledge about the process, the dependencies between the different processes, and the mean time of the tasks. While the output is, above all, the design, and the best distribution of the processes in the plant.

The knowledge base system receives expert knowledge coded into facts, rules, heuristics, and procedures. The expert system comprises three elements including database, knowledge base, and inference engine. The general architecture of the knowledge database and expert system is presented in [29]. The database possesses information in terms of fact or heuristics based on user interest of a specific problem domain. Knowledge base possesses domain knowledge expressed in terms of mathematical logic.

There are several ways to represent the relationships between classes, attributes, and class procedures. This work uses Unified Modelling Language (UML) due to its simplicity, intuitive use, and extensive documentation [30]. Figure 3 represents the class diagram of the conceptual reference framework where the process area of the diagram was described in six main process categories: Upper pieces cutting, Manufacturing, Quality test, Flash cutting, Cleaning/Polishing, and Packaging. Such classification was detailed following the formalisation proposed by [31,32]. This study is focused on the manufacturing process. The other processes, as they are not part of the manufacturing process, will not be part of the design in the current study. Each process category is subclassified in the main subprocess described in the following section in the study. The subprocess are as follows: sewing,

leather buffing, pieces pre-shaping, cementing, lasting, milling, drilling, and grinding. These sub-processes will be studied in depth in the research. It should be noted that the procedure is valid for the design of any type of shoe production factory, independently of the used production process: direct-injection technique or mounting-and-cementing technique. This project considered the knowledge provided by experts to generate a UML model that can be extrapolated to any type of conventional shoe manufacturing plant and for any factory size.

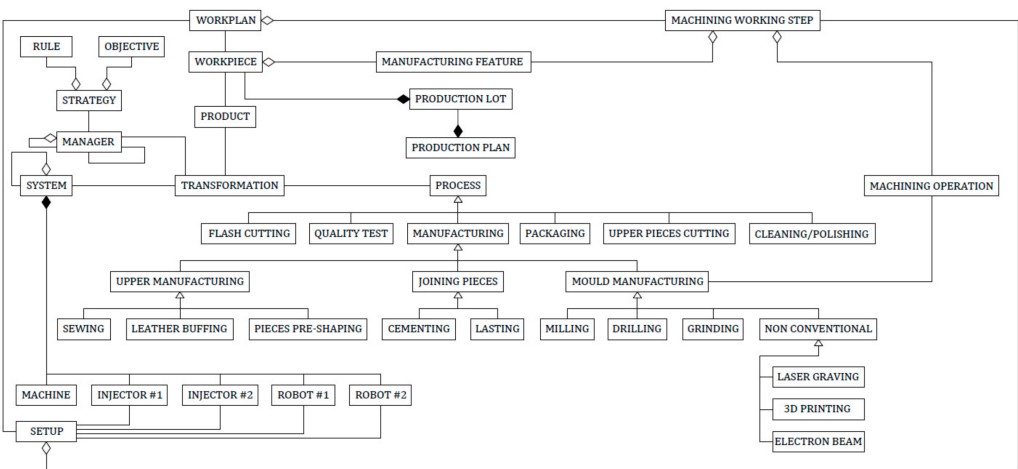

**Figure 3.** Class diagram of the conceptual reference framework.

## 4.2. Software Simulation Approach

Every simulation software has its upsides and downsides, and the choice may be made depending on the application. Probably, Promodel [33] and Arena [34] are the most traditional ones and led the way to the development of these technologies as they gave the opportunity of modelling and simulating a variety of systems by using logical blocks. Thereby, Simio [35] and Flexsim [36] set themselves as the evolution of these tools by implementing multiagent methods, better graphic simulation quality, and even increasing the ease of use for inexperienced users. Nevertheless, they show some limitations when it comes to implementing more complex routines. As it has been mentioned, there are many software tools that allow modelling and simulation of discrete event processes. Some of them are useful in a very generic way and others are specifically oriented to some types of systems and models. In this sense, Anylogic [37] provides a Process Modelling Library which is a collection of objects that allows to define process works and their associated resources in order to conduct discrete event simulation properly. The AnyLogic simulation software has proven to be a useful tool for solving facility design problems. In [38] AnyLogic is applied to evaluate the impacts of supply chain efficiency on coordinated and uncoordinated production and order control policies. The authors of [39] studied Tesla's six-link closed-loop supply chain from the resilience perspective using disruption and recovery scenarios. The authors used the simulation software AnyLogistix, which is similar in function to the AnyLogic simulation software platform. Furthermore, Process Modelling Library works closely with the presentation framework and enables to develop 3D process animations and thereby facilitate the detection of errors during simulation. In addition, Anylogic has a free edition called Personal Learning Edition. This edition can be used without time constraints, but it puts some limits on the number of agents and blocks. This free of charge edition has been key to decide using Anylogic in this work. Results presented in this article can be replicated without having specific software and especially important, this methodology can be used on any other shoe plant design. All in all, to accomplish the purpose of this paper, it was necessary to start from scratch and define every characteristic of tackled models. On the one hand, it is necessary to model the logical blocks that will define the varied operations implicated in both manufacturing

systems. On the other hand, in order to make the simulation as close to reality as possible, it is very useful to define the physical layout of the systems and configure the relationship between logical and physical design, defining the 2D or 3D animation of the models and then making concepts and ideas to be more easily verified.

### 4.3. Time Estimation for Process Operations

Both processes have many operations in common. Especially, those related with the upper are exactly the same in both production technologies. They differ on the bonding process between the outsole and the upper. Table 1 shows the times per product for both systems. It is easy to observe that the direct-injection process has less operations, saving time in the global process. For the estimation of the operation times of the processes, we were assisted by the company Simplicity Works Europe SL (Elche, Spain) [40]. This company is a manufacturing technology company devoted to several industrial areas. One of these areas is the footwear production, where the company has very relevant international clients. Its business activity requires a deep knowledge of industrial manufacturing processes, which allows in the case of footwear production, to know the set of processes, production partial times, and manufacturing flows. This in-depth knowledge is part of the business company and allows the optimisation of resources for the installation of new industrial production plants.

**Table 1.** Features of optimal sequence for Manufacturer 1 and manufacturing times per product. This information is associated with direct-injection and the mounting-and-cementing process.

| Task | Direct Injection Mounting and Cementing Time per Shoe (s) | |
| --- | --- | --- |
| Material placement and projections adjustment | 6.5 | |
| Cutting | 39 | |
| Upper marking | 30 | |
| Tongue marking | 9 | |
| Milling | 30 | |
| Temporary unions | 45 | |
| Toecap sewing | 37 | |
| Quarter sewing | 67 | |
| Heel sewing | 37 | |
| Gluing between tongue, upper, and lining | 52 | |
| Gluing between upper and lining | 47 | |
| Gluing foam in tongue | 22 | |
| Top moulder | 52 | |
| Heel moulder | 52 | |
| Eyelets | 36 | |
| Top mounting | - | 17 |
| Heel and shank mounting | - | 17 |
| Marking and brushing | - | 26 |
| Gluing sole and upper | - | 40 |
| Drying and glue activation | - | 20 |
| Press outsole bonding | - | 38 |
| Flash cutting | 10 | - |
| Last removal | 6 | |
| Cleaning | 13 | |
| Insole and shoelace | 51 | |
| Packing | 14 | |
| Cutting | 39 | - |

## 5. Case Study 1: The Direct-Injection Process

The first case is studied is the direct-injection process. There are a lot of tasks in this process which have to be completed in a determinate order. This process is developed with aim to reach a high level of automatisation, therefore it seems coherent that the correct

position of the machinery and a correct design could save some minutes in the overall time of the process.

### 5.1. Time Estimation for Process Operations

In order to model any manufacturing system, it is necessary to define the number of operations as well as its respective average times. In addition, mean features of the process must be defined. There are several types of rotary injection machines with different numbers of stations and injectors. In this case, the features of the rotary machine considered are described in Table 2.

**Table 2.** Rotary machine features (direct-injection process).

| | |
|---|---|
| Number of stations | 24 |
| Cycle of time (s) | 16 |
| Movement time between stations (s) | 2 |
| Number of injectors | 2 |

In this sense, the agents and parameters that define the model in Anylogic are configured. Every operation time, speed, and interval delivery are represented by a parameter (see Figure 4a). The model can be more easily adjusted when it comes to finding the optimum balance of the system. Regarding the agents (see Figure 4b), the main resources of the process such as 'Worker', 'Truck', 'Part', and 'Pallet' are defined.

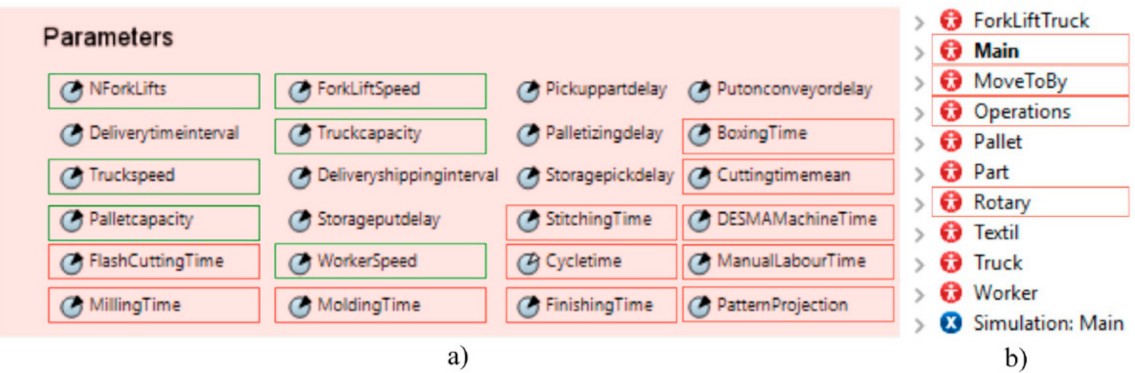

**Figure 4.** (**a**) System parameters (with a green frame the parameters that represent agent characteristics are outlined while parameters framed in red represent the main times of the system). This representation is associated with the direct-injection process. (**b**) Agents denned (agents framed in red are necessary in order to store logical blocks). These agents are associated with the direct-injection process.

### 5.2. Modelling Results

Once the agents and parameters are established, the system can be modelled. It is possible to differentiate several phases of the modelling process, that are listed below:

- Delivery, storage, and shipping of materials and products: First of all, a system for the delivery and storage of raw material is modelled. Either the storage capacity or the interval delivery time can be controlled through the parameters that have been previously defined in order to avoid stock shortage.
- Limiting factor: Secondly, the 'Rotary' agent that represents the rotary injection machine is configured. Simultaneously, the relationship between the rotary machine and the rest of the system is defined. Figure 5 shows the logical blocks used for modelling the rotary injection machine movement.
- Pre limiting factor operations: After having properly defined the limiting factor, the operations prior to that point must be modelled and the number of resources needed to keep the rotary machine at full capacity established.

- Post limiting factor: Finally, the operations of finishing and packing are modelled. At this moment, it is necessary to establish the number of resources needed to extract every product injected from the rotary injection machine without getting new queues.

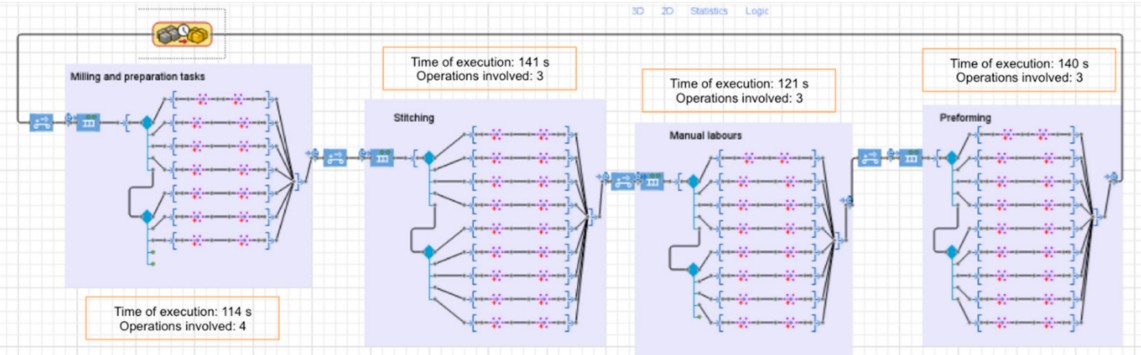

**Figure 5.** Logical blocks in 'Rotary' agent and configuration of the transition between the two phases. These blocks are associated with the direct-injection process.

In order to get an as simple as possible model, tasks that are similar to each other are grouped in an only time parameter. The logical blocks that will be run during the simulation are shown in Figure 6. Furthermore, Figure 7 shows the spatial representation of the final model in 2D. It is important to outline some of the results provided by the simulations once the model is finished and optimised with the rotary injection machine working at full capacity in every moment. With the resources employed, we can ensure that the only bottleneck of the system is the limiting factor and the product spends 45.925 min on average from the beginning to the end of the system. After simulating 8 working hours, 787 pairs of shoes were produced and only 11 products remained waiting to be placed in the rotary injection machine which would take only 13.017 min to be completely finished.

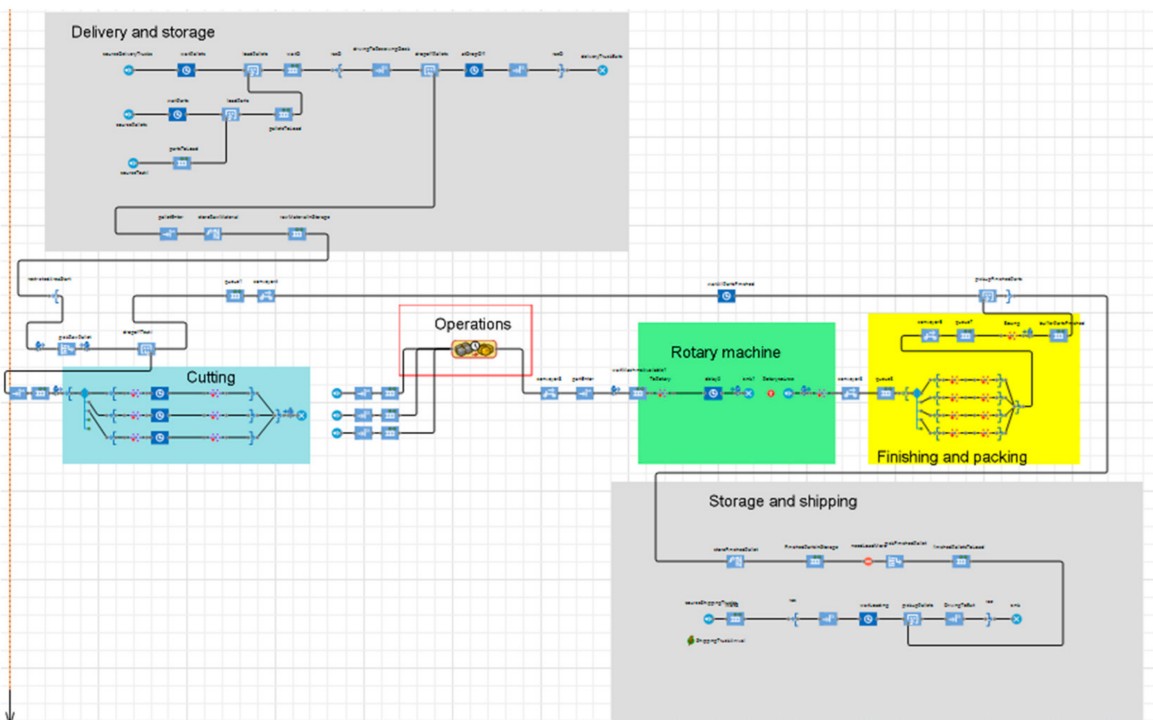

**Figure 6.** Block diagram of the direct-injection process.

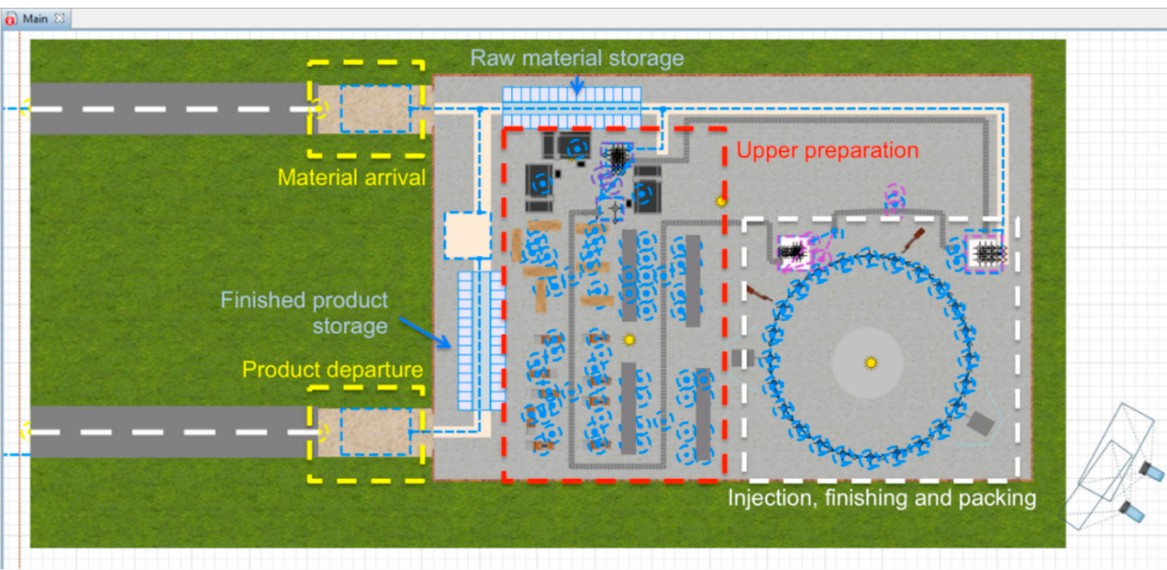

**Figure 7.** Spatial representation of the direct-injection system.

## 6. Case Study 2: The Mounting-and-Cementing Process

The mounting-and-cementing process has many operations in common with the direct-injection one. Especially, those related with the upper are exactly the same in both production technologies. They differ on the bonding process between the outsole and the upper. In this sense, the direct-injection model explained in the previous section is used to introduce this manufacturing process increasing the quality of the final product. In this case, as it was done with the direct-injection process, it is necessary to identify the limiting factor of the system. The main activity in this case is the mounting and cementing of the outsole in order to compare this technique with the direct-injection one. In general, medium size factories do not have more than two mounting-and-cementing machines. Even if it would be possible to improve the productivity of a mounting-and-cementing system by increasing the number of stations per task, it is common to consider the mounting machines as the limiting factor. In this sense, it is interesting to optimise the system considering the mounting machines as the bottleneck of the process that has to work at full capacity during the whole simulation.

### 6.1. Input Data

In order to model the mounting-and-cementing process, the operations involved in the process as well as its average times must be known. Based on this information, parameters and agents needed for properly modelling this process in exactly the same way as it was previously done for the direct-injection case are defined.

### 6.2. Modelling Results

To define the logical and spatial aspects of mounting-and-cementing process, common points with direct-injection model have been reused. In fact, operations of delivery, storage, and shipping are exactly the same. Some operations are common in both methods and therefore, the system is modelled following the same four steps of the previous section. The logical blocks of the processes that are modelled in this case replacing the rotary machine of the direct-injection system by two mounting-and-cementing machines are shown in Figure 8. Every single step of the mounting-and-cementing process was modelled in order to maintain the limiting factor of the system at full capacity, thus forming an only bottleneck on the model at the entrance of the mounting machines.

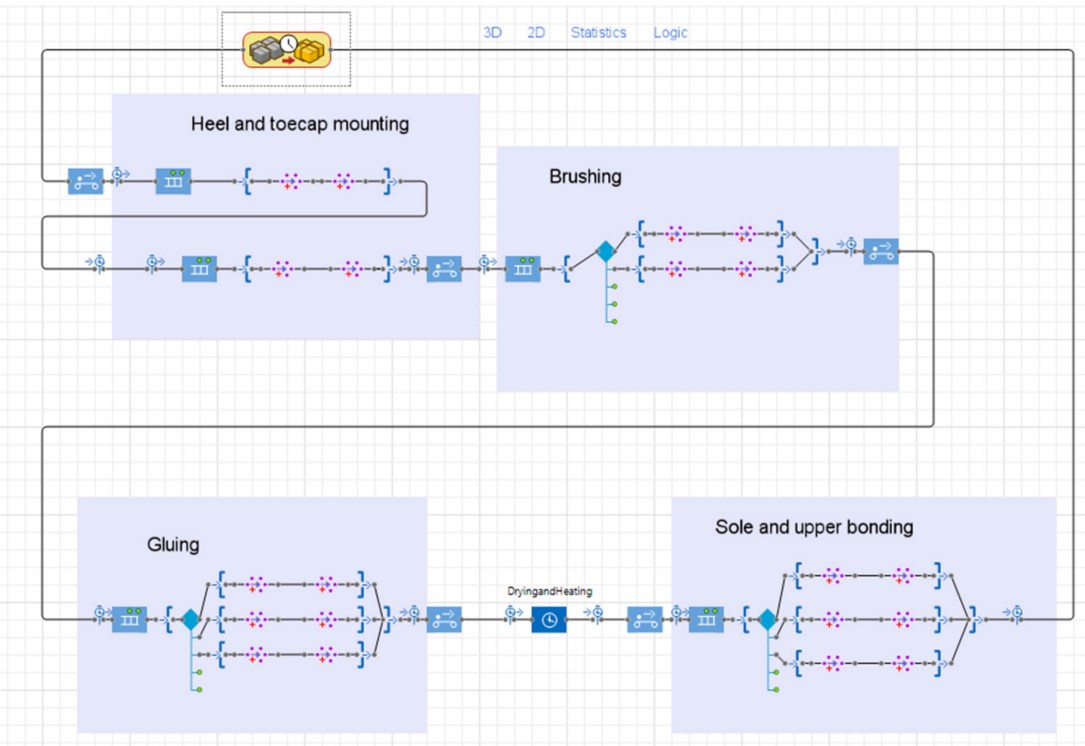

**Figure 8.** Logical blocks of the characteristic operations of the mounting-and-cementing process.

Figure 9 shows the spatial representation of the model in two dimensions. To conclude, the system is optimised with only one bottleneck (first mounting-and-cementing machine) that contains 24 agents (shoes) unfinished at the end of a 8 h of simulation which would take only 12.381 min to be completely finished. Moreover, 660 pairs of shoes would be manufactured at the end of the day and each product would take on average 47.275 min to be manufactured.

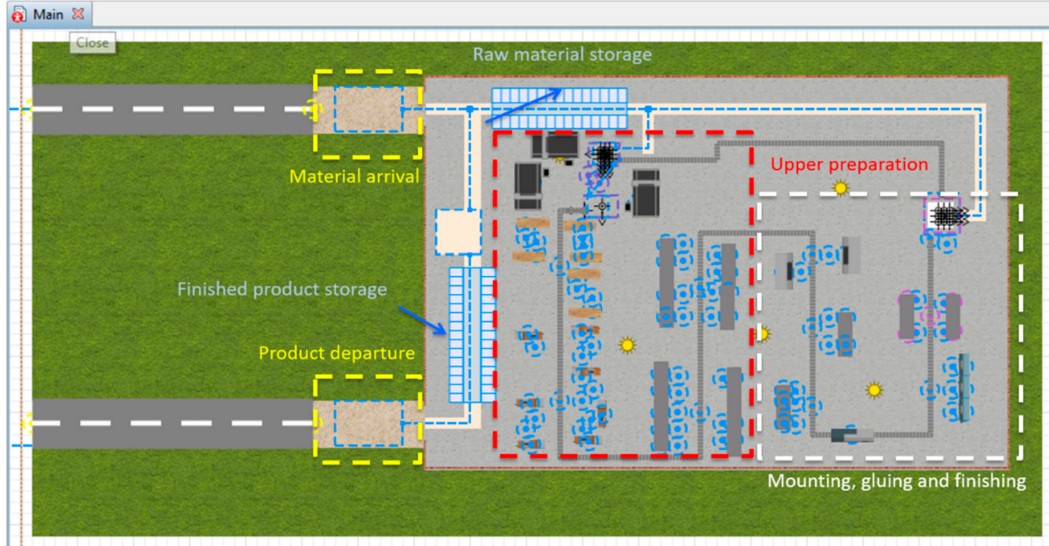

**Figure 9.** Spatial representation of the mounting-and-cementing process.

## 7. Comparison between Both Processes

Once the simulations were carried out using an expert system for both processes, they were compared taking into account factors such as time per product, efficiency, and the number of workers, thus obtaining more analytical data on the plant design. Table 3 shows how every operation (after the cutting process) is considered for both processes, the number of workstations modelled for each group of tasks in order to keep the limiting factor at full capacity, and the average efficiency of the workers.

**Table 3.** Summary of modelled processes. This information is associated with the direct-injection and the mounting-and-cementing process.

| Process | Operations | Time per Product | | N Workers | | Efficiency | |
|---|---|---|---|---|---|---|---|
| | | Operation Time | Total Time | DI | M&C | DI | M&C |
| Milling and preparations tasks | Upper marking | 30 | | | | | |
| | Tongue marking | 9 | | | | | |
| | Milling | 30 | 114 | 7 | 6 | 88.57% | 86.67% |
| | Temporary unions | 45 | | | | | |
| Stitching | Toecap sewing | 37 | | | | | |
| | Quarter sewing | 67 | | | | | |
| | Heel sewing | 37 | 141 | 9 | 8 | 83.93% | 82.21% |
| | Gluing tongue, upper, and lining | 52 | | | | | |
| Manual labours | Gluing upper and lining | 47 | | | | | |
| | Gluing foam | 22 | 121 | 8 | 7 | 81.70% | 80.60% |
| | Top moulder | 52 | | | | | |
| | Heel moulder | 52 | | | | | |
| Preshaping | Eyelets | 36 | 140 | 9 | 8 | 81.96% | 79.04% |
| | Last removal | 6 | | | | | |
| Finishing | Cleaning | 13 | 70 | 4 | 4 | 88.00% | 78.00% |
| | Insole and shoelace | 51 | | | | | |

The processes that are shown in the Table 3 together with the cutting, packing, and entry/exit in the rotary injection machine operations involve a total of 37 workers for direct injection and 33 workers for mounting and cementing to allow the proper development of the system.

Table 4 shows the summary of the resources needed for the mounting-and-cementing process in order to accomplish the purpose of optimisation and efficiency of resources, these operations are additional in this process and are not needed in the direct injection. Therefore 10 workers are needed. Considering the operations shown in the tables together with the cutting and packing, an amount of 37 workers for direct injection and 43 workers for mounting and cementing are required.

**Table 4.** Features of optimal sequence for Manufacturer 1 and summary of characteristic operations of the mounting-and-cementing modelled.

| Process | Time per Product (s) | N of Workers | Efficiency |
|---|---|---|---|
| Toecap mounting | 17 | 1 | 89.00% |
| Heel and shank mount | 17 | 1 | 86.00% |
| Brushing and marking | 26 | 2 | 67.50% |
| Gluing sole and upper | 40 | 3 | 76.00% |
| Press outsole bounding | 38 | 3 | 61.66% |

These results are coherent and in line with expected data. The direct injection process is characterised by a higher production volume with fewer workers, which means higher productivity per worker and work shift. On the other hand, the mounting and cementing process involves a lower initial investment in machinery and although the system is less

efficient, it allows flexibility in production by adapting the number of workers. In other words, direct injection is significantly more profitable if production is close to the maximum, but for lower productions, profitability tends to be similar in both cases and the return on investment (ROI) grows. In this case, the mounting and cementing process is more attractive for producers.

## 8. Conclusions

This article presented an expert system to support production engineers in the design and dimensioning of a shoe production plant. The expert system proposed in this research consists of a set of rules, an object-oriented modelling, technical information of components, and procedural functions. This project was focused on optimising a shoe production plant based on an expert system and carrying out a series of iterations to adjust the production resources (human and material) following a trial-and-error process and using a simulation tool that provides the following advantages:

- Proposal of productive design alternatives: resources and productivity of both analysed methods (direct-injection and mounting-and-cementing).
- Industrial process design when exact or complete data does not exist.
- Space needs for different resources, especially for production and storage (finished product and raw materials).

The proposed approach enables to get relatively accurate results in a less time consuming and expensive way than testing alternatives on a real production plant. Important information that can be extracted from this work is the number of shoes produced per worker during a labour day (one shift and 8 working hours was considered). Table 5 shows that the productivity of a worker in a direct-injection factory is 36.60 shoes/day/shift meanwhile the productivity of a worker in a mounting-and-cementing factory is 28.08 shoes/day/shift. That means that a direct-injection factory is a 30.34% more productive than a mounting-and-cementing one. In addition, the procedure shown in this article is scalable and can be extrapolated to other cases. It can be easily applied to bigger factories, where the warehouse dimensioning is essential and time from a customer order to the delivery is very valuable information. However, what maybe marks the difference is the possibility to have information about the number and size of lasts and moulds required for a certain production. These values are usually estimated based on the previous knowledge of the managing team, but mistakes on getting the number of moulds and lasts generate losses that impact on the company's benefit.

**Table 5.** Productivity compression of a factory with direct-injection vs. mounting-and-cementing systems.

| | One Shift Labour Day (8 h) | N of Workers | Productivity Ratio Shoes per Worker per Day |
|---|---|---|---|
| Direct injection | 1574 pairs of shoes | 37 | 36.60 |
| Mounting-and-cementing | 1320 pairs of shoes | 43 | 28.08 |

The content of this article shows the results of the research work carried out as a collaboration between SimplicityWorks company and Miguel Hernandez University. The results obtained will be used by the company to design its clients' shoe manufacturing plants. Simplicity Works has validated task and sub-task times and plans to use this knowledge in its customer projects. The criteria and tools used in this study, especially the expert system developed, will help to economically analyse the investment in equipment and human resources of footwear producers.

The next step in the study process of the present work and as future lines, is the generalisation of the proposed methodology to other types of manufacturing or production plants and to check the robustness of the methodology presented in other more complex environments, when the full process is automatised with robotics systems presenting new dependencies.

**Author Contributions:** Conceptualization J.B.M. and D.C.; methodology, F.N.; software J.B.M.; validation, C.P.-V. and J.V.S.-H.; formal analysis J.B.M.; resources, D.C.; writing—original draft preparation, J.B.M.; writing—review and editing, C.P.-V.; supervision, J.V.S.-H.; project administration F.N. All authors have read and agreed to the published version of the manuscript.

**Funding:** This research was funded by Ministerio de Economia y Competitividad of the Spanish government grant number [RTC-2016-5408-6] and Generalitat Valenciana (financial support through grant PROMETEO/2021/063).

**Institutional Review Board Statement:** Not applicable.

**Informed Consent Statement:** Informed consent was obtained from all subjects involved in the study.

**Data Availability Statement:** This study does not report any data.

**Acknowledgments:** This research has been supported by DESMA Schuhmaschinen GmbH (financial and technical support) SimplicityWorks Europe (technical support), the Ministerio de Economia y Competitividad of the Spanish government (financial support through grant RTC-2016-5408-6) and Generalitat Valenciana (financial support through grant PROMETEO/2021/063).

**Conflicts of Interest:** The authors declare no conflict of interest.

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
