# Peer review of "Conceptual and Preliminary Design of a Shoe Manufacturing Plant"

_applsci, doi:10.3390/app112211055_

Round 1
Reviewer 1 Report
The authors present an expert system that supports production engineers in the design and dimensioning of a shoe production plant. The goal is to optimize the process. Two methods are analysed, namely direct-injection and mounting-and-cementing. The results are coherent and in line with expected data.
The article is clearly structured and well-written in general (also from the point of view of the English language).The modelling process is made with rigour and can be extrapolated to any type of conventional shoe manufacturing plant and for any factory size.
The article is original and has a high degree of novelty. This is also proven by Turnitin, a software that detected a percentage of plagiarism of only 13% (with the option "Exclude Bibliography").

Author Response
To whom it may concern,
We would like to thank the reviewers for their comments because they have helped us a lot to improve the submitted paper.
We have made several changes to the paper to improve the structure and compression of the paper.
In this way we believe that the paper improves its contribution to knowledge.
Reviewer 2 Report
In this article, the authors try to design a method to help in the design and optimization of footwear production plans based on an expert system. After introducing it, without any detailed and founded explanation of the architecture of the system, two case are studied, in which, I was not able to see the application of the expert system and to understand how this system can help in the design and optimization of the plant design.
For these reasons, and all the comments present below, I honestly believe that this paper must be rejected. It has many flaws and problems which must be solved and deeply studied.
Main reasons
- What is the difference in the design of a shoe manufacturing plant and the design of another manufacturing plant? In the lines 28 – 29 it is said: ‘The position of the machines and the different processes steps can reduce the time of the whole process, and this is little studied in the literature’. This is a common and well-known problem when a manufacturing plant is designed. Why do the authors say that is little studied in the literature?
- In lines 36 – 37, the authors said: ‘For this reason, this research presents a more scientist method to design a manufacturing plant combined different elements such as expert systems and simulations. Therefore, it seems coherent to look for the solution in an expert system.’ What do the authors mean when they say a more scientist method? Why did the authors decide to use an expert system and no other approach?
- In lines 39 – 42, authors said: ‘These rules are mainly obtained from technical bibliography and expert knowledge. There are several techniques for acquiring knowledge for an expert system, such as those presented by A. Kidd [1].’ Which of the techniques for building the knowledge base of the Expert System, of those which are presented in the reference [1], did the authors use?
- In lines 47 – 49, authors said: ‘This work presents an expert system to support engineers in the conceptual and preliminary design phases of a shoe production factory, considering human and material resources and real processing times.’ Do the authors consider that an expert system is the same than a decision support system? I suggest the authors to review the paper of Pfeifer and Lüthi [1].
[1] Pfeifer, R.; Lüthi, H.-J. Decision Support Systems and Expert Systems: A Complementary Relationship? In Expert Systems and Artificial Intelligence in Decision Support Systems; Springer: Dordrecht, The Netherlands, 1987; pp. 41–51.
- Which design methodology are the authors following in the design process of the plant?
- In the Section 2 a literature review of some applications of expert systems in the industry is performed. Authors present some works without describing them in detail. Authors should add some lines about every work. Furthermore, some of the approaches presented do not use expert systems, so authors must point that, for example: ‘The work presented by [16] proposes a scheduling module focused on the short term in order to respond quickly to market needs and changes in a flexible manner. The module is composed of a finite capacity scheduler integrated with a new software based on the Analytical Hierarchy Process (AHP) decision support system. This module considers the aspects related to order importance as complexity or urgency and assign each order a priority.’ Please review the whole section. I consider also that the authors should make a table where they summarize all the works, one by one, as it will be easier for the reader. Authors should also add some lines where they describe the conclusions obtained after the review, explaining the general trends, when expert systems are used in the industry plant design.
- In lines 90 – 93, it is said: ‘The last years, with the growth up of the artificial intelligence, it is combined with expert systems in order to improve the solutions in [7], [8] and [9] an artificial intelligence to improve some fields such as chemical process, food or agriculture is presented.’ Actually, expert systems are an artificial intelligence approach.
- In lines 225-226, authors said: ‘The research is based on the combination between an expert system and a model simulation.’ Why is it based on that combination? What do the authors understand as a model simulation?
- Authors must add a section where they present the technical grounds of the work and where they answer to several questions:
- What is it an expert system? What are the components of an expert system?
- Do the authors propose a decision support system? It that’s the case, what is a decision support system?
- The model presented in this manuscript needs to be considered within the information systems framework. Maybe the presented model and its implications could be understood as an artefact according to the design science research. In order to validate the contributions in the field of information systems, it is necessary that the authors evaluate their proposal according to the guidelines and principles proposed by Hevner et al. [2, 3].
[2] Hevner, A.R.; March, S.T.; Park, J.; Ram, S. Design science in information systems research. MIS Q. Manag. Inf. Syst. 2004, 28, 75–105, doi:10.2307/25148625.
[3] Hevner, A.R.; Chatterjee, S. Design Research in Information Systems: Theory and Practice; Springer: New York, NY, USA, 2010; ISBN 978-1-4419-6107-5.
- Sections 4.1 and 4.2 have the same title, Results and discussion. Why? Shouldn’t the method be presented before the results and discussion?
- Figure 3 must be explained in more detail, explaining each of the decisions made to build the diagram, and the information flow. How was the design of the model carried out?
- Which type of inference engine does the expert system use? Which type of inference mechanism does it apply?
- Does de expert system proposed by the authors manage the uncertainty? If that’s the case, how?
- The authors must define which are the inputs and the outputs of the proposed expert system. Authors must also present a subset of rules.
- In Section 4.3, authors describe several simulation software; Why? Is that necessary?
- After analysing both case studies, in what is the expert system helping? What is the output of the system? How is the expert system helping to design or to optimize the layout and reduce the time process?
- A discussion section must be added. What is the difference between this proposal and other? Why is this proposal better? What are the weaknesses of the system?
- What are the future lines of work?
Other comments
- The abstract should be rewritten, as it does not provide the global idea of the manuscript. The methods and the conclusions should be described deeply in the abstract.
- I consider that the organization of the sections should be rethought. Maybe, section 3, should be before than section 2.
- Authors should pay attention to the text:
- In lines 26 – 26 it is said: ‘The present design of a manufacturing plant is an important feature, more in the current days in which the automation is an upward trend’. Which plant? That is the first line of the introduction section. The problem should be presented properly before.
- References must be reviewed. Many details are missing in books and articles.
- When a reference is cited in the paper, authors should introduce it properly. I suggest saying: ‘In the paper of someone [], it is …’, instead of: In []…; For example, in Line 115, it is said: ‘[13] is aimed…’
- In Line 113, it is said: ‘In chapter 10…’; See also line 116, where it is said: ‘Chapter 8 …’… What do you mean?
- In lines 142 – 143, it is said: ‘Casual shoes, one of the segments analyzed and sized in R. Reddy (2019)…’. Is a citation missing?
- The ‘1. Subsection’ must be renamed.
- Can the authors use Figure 1? Does it belong to a third party? Also with Figure 2 (despite saying Courtesy of DESMA Schuh-maschinen GmbH), do the authors have the permission to use that figure?.
- In line 256 the authors use the acronym UML. What is the meaning?

Author Response
To whom it may concern,
We would like to thank the reviewers for their comments because they have helped us a lot to improve the submitted paper.
Reviewer: What is the difference in the design of a shoe manufacturing plant and the design of another manufacturing plant? In the lines 28 – 29 it is said: ‘The position of the machines and the different processes steps can reduce the time of the whole process, and this is little studied in the literature’. This is a common and well-known problem when a manufacturing plant is designed. Why do the authors say that is little studied in the literature?
Author: Production plant optimization is a well-known area of knowledge in literature. It includes optimization of resources and layout design (distribution of elements along the factory). When we refer to this research as little studied, we are only talking about the shoe industry. This article applies known techniques to a non exploited industrial sector. It has been clarified on the text.
R: In lines 36 – 37, the authors said: ‘For this reason, this research presents a more scientist method to design a manufacturing plant combined different elements such as expert systems and simulations. Therefore, it seems coherent to look for the solution in an expert system.’ What do the authors mean when they say a more scientist method? Why did the authors decide to use an expert system and no other approach?
A: Shoe factory design is usually made by experts that have been done it before. It is common to adjust some parameters using a trial and error technique over the real factory. This article presents a medium-size production plant where all elements are properly dimensioned (optimized). Here, a reader can find which machines will he need, how many workers should the factory have or how many pairs of shoes will be produced per day. The introduction has been slightly modified and Conclusions section.
R: In lines 39 – 42, authors said: ‘These rules are mainly obtained from technical bibliography and expert knowledge. There are several techniques for acquiring knowledge for an expert system, such as those presented by A. Kidd [1].’ Which of the techniques for building the knowledge base of the Expert System, of those which are presented in the reference [1], did the authors use?
A: This text has been rewritten to clarify and specify some of the relevant features presented by Kidd and used in this research:
“There are several features such as flow description, relationship and dependencies be-tween processes, task times (including statistical distribution) or utilization rate that are required to build the Expert System. All of them are presented by A. Kidd in [1].”
R: In lines 47 – 49, authors said: ‘This work presents an expert system to support engineers in the conceptual and preliminary design phases of a shoe production factory, considering human and material resources and real processing times.’ Do the authors consider that an expert system is the same than a decision support system? I suggest the authors to review the paper of Pfeifer and Lüthi [1]. [1] Pfeifer, R.; Lüthi, H.-J. Decision Support Systems and Expert Systems: A Complementary Relationship? In Expert Systems and Artificial Intelligence in Decision Support Systems; Springer: Dordrecht, The Netherlands, 1987; pp. 41–51.
A: Thank you a lot for the recommendation, Pfeifer and Lüthi paper seems really interesting. We agree that this sentence is inaccurate and a new writing is proposed:
“This work presents an expert system, generated using the knowledge of an experienced group of people and where human and material resources and real processing times have been considered.”
R: Which design methodology are the authors following in the design process of the plant?
A: This paper presents an expert system following a Function Oriented Manufacturing Systems (FOMS). Product Oriented Manufacturing System (POMS) is an alternative to FOMS but after presenting both methodologies to our expert board, we decided to follow a FOMS.
R: In the Section 2 a literature review of some applications of expert systems in the industry is performed. Authors present some works without describing them in detail. Authors should add some lines about every work. Furthermore, some of the approaches presented do not use expert systems, so authors must point that, for example: ‘The work presented by [16] proposes a scheduling module focused on the short term in order to respond quickly to market needs and changes in a flexible manner. The module is composed of a finite capacity scheduler integrated with a new software based on the Analytical Hierarchy Process (AHP) decision support system. This module considers the aspects related to order importance as complexity or urgency and assign each order a priority.’ Please review the whole section. I consider also that the authors should make a table where they summarize all the works, one by one, as it will be easier for the reader. Authors should also add some lines where they describe the conclusions obtained after the review, explaining the general trends, when expert systems are used in the industry plant design.
- The addition of a new table to list and summarize references used on the introduction is a good idea. However, we have slightly modified the text to explain references of Section 2 in more detail.
R: In lines 90 – 93, it is said: ‘The last years, with the growth up of the artificial intelligence, it is combined with expert systems in order to improve the solutions in [7], [8] and [9] an artificial intelligence to improve some fields such as chemical process, food or agriculture is presented.’ Actually, expert systems are an artificial intelligence approach.
A: The text has been changed to be clearer and closer to the stated approach:
“The last years, with the growth up of the artificial intelligence, the expert systems are considering an approach of the artificial intelligence and they are use in order to improve their solutions in [17], [18] and [19].”
R: In lines 225-226, authors said: ‘The research is based on the combination between an expert system and a model simulation.’ Why is it based on that combination? What do the authors understand as a model simulation?
A: This paragraph has been rewritten to clarify that a simulation model is not used but a simulation process to validate the expert system proposed:
This research is based on an expert system design supported by discrete event simulation. Firstly, the expert knowledge collected is essential in order to create a data base. This data base will be used by the expert system to develop the design required. In this research a shoe manufacturing plant is designed using both tools. Secondly, a simulation of the production plant with it will be used to verify the solution. On the other hand, a high number of accurate information about the solution (task times, number of workers, etc.) will be gathered in the simulator.
R: Authors must add a section where they present the technical grounds of the work and where they answer to several questions:
- What is it an expert system? What are the components of an expert system?
- Do the authors propose a decision support system? It that’s the case, what is a decision support system?
- The model presented in this manuscript needs to be considered within the information systems framework. Maybe the presented model and its implications could be understood as an artefact according to the design science research. In order to validate the contributions in the field of information systems, it is necessary that the authors evaluate their proposal according to the guidelines and principles proposed by Hevner et al. [2, 3]. [2] Hevner, A.R.; March, S.T.; Park, J.; Ram, S. Design science in information systems research. MIS Q. Manag. Inf. Syst. 2004, 28, 75–105, doi:10.2307/25148625. [3] Hevner, A.R.; Chatterjee, S. Design Research in Information Systems: Theory and Practice; Springer: New York, NY, USA, 2010; ISBN 978-1-4419-6107-5.
A: References provided are really interesting. Authors have overseen them to understand reviewer’s comment and some lines have been added to the text. References in the article to a decision support system has been removed to avoid misunderstandings. This research presents an expert system based on knowledge of experienced shoe factory managers. Simulations are used to validate the expert system and to show results of two use cases. These ideas have been introduced in several points of the manuscript.
About the components of the expert system, Figure 3 shows the diagram of elements and its connections.
R: Sections 4.1 and 4.2 have the same title, Results and discussion. Why? Shouldn’t the method be presented before the results and discussion?
This mistake has been corrected.
R: Figure 3 must be explained in more detail, explaining each of the decisions made to build the diagram, and the information flow. How was the design of the model carried out?
A: Text has been added to better clarify the figure:
“Figure 3 represents the class diagram of the conceptual reference framework where the process area of the diagram has been described in six main process categories: Upper pieces cutting, Manufacturing, Quality test, Flash cutting, Cleaning/Polishing and Packaging. Such classification has been detailed following the formalization proposed by [30] and [31]. Each process categories are subclassified in the main subprocess describe in the follows section in the study. This subprocess are: sewing, leather buffing, pieces pre-shaping, cementing, lasting, milling, drilling and grinding. These sub-processes will be studied in depth in the research.”
R: What are the future lines of work?
A: A paragraph is added to present the future lines:
“The next step in the study process of the present work and as future lines, is the generalization of the proposed methodology to other types of manufacturing or production plants and to check the robustness of the methodology presented in other more complex environments.”
Other comments
R: The abstract should be rewritten, as it does not provide the global idea of the manuscript. The methods and the conclusions should be described deeply in the abstract.
A: The abstract has been rewritten in order to improve it.
R: I consider that the organization of the sections should be rethought. Maybe, section 3, should be before than section 2.
A: The advice has been taken into account.
R: When a reference is cited in the paper, authors should introduce it properly. I suggest saying: ‘In the paper of someone [], it is …’, instead of: In []…; For example, in Line 115, it is said: ‘[13] is aimed…’
A: The advice has been taken into account and some references haven been changed.
R: In Line 113, it is said: ‘In chapter 10…’; See also line 116, where it is said: ‘Chapter 8 …’… What do you mean?
A: This was a mistake, it has been corrected.
R: The ‘3.1. Subsection’ must be renamed.
A: The subsection has been renamed.
R: Can the authors use Figure 1? Does it belong to a third party? Also with Figure 2 (despite saying Courtesy of DESMA Schuh-maschinen GmbH), do the authors have the permission to use that figure?.
A: Yes, we have the permission of DESMA.
R: In line 256 the authors use the acronym UML. What is the meaning?
A: The acronym has been defined:
“Unified Modeling Language (UML)”
Reviewer 3 Report
I see that the manuscript is a good contribution in thr manufacturing are.
It presents a genuine idea supporting its importance.
For me as a reader I was surprised to learn some new info at every time in the work.
However i believe the structure if the work needs a double check
In section 4, we can see two sections 4.1 results and discussions and i believe this had to do with the experimental part of the manuscript.
The results are well shown as result are clear and straightforward.
Author Response
To whom it may concern,
We would like to thank the reviewers for their comments because they have helped us a lot to improve the submitted paper.
Reviwer: However i believe the structure if the work needs a double check
We have made several changes to the paper to improve the structure and compression of the paper.
In this way we believe that the paper improves its contribution to knowledge.
R: In section 4, we can see two sections 4.1 results and discussions and i believe this had to do with the experimental part of the manuscript.
This error has been corrected and the structure of the section has been well established.
Reviewer 4 Report
This paper proposed a procedure to design footwear production plants by using a simulation approach. The topic is interesting and satisfies the scopes of this journal. The introduction of this studied problem should be modified to make it more clear. The proposed method also needs to be rewritten to make it easier to understand.
Author Response
To whom it may concern,
We would like to thank the reviewers for their comments because they have helped us a lot to improve the submitted paper.
Reviwer: The introduction of this studied problem should be modified to make it more clear. The proposed method also needs to be rewritten to make it easier to understand.
We have made several changes to the paper to improve the structure and compression of the paper.
In this way we believe that the paper improves its contribution to knowledge.
Round 2
Reviewer 2 Report
The authors have performed a very superficial revision of their manuscript. They didn’t reply to all my comments (avoiding those which are more technical, and related with the definition of an expert system, or just replying to them without answering to what was pointed). The manuscript present serious flaws and problems, not only from the design perspective, but also from the conceptual one, being far away for being considered for publication. Moreover, it was really difficult to perform the review, as the authors did not provide the Lines of their changes on each of the comments.
For these reasons, this manuscript must be rejected again. I consider that this manuscript will not improve even with a Major Revision.
Below, it is possible to find some comments that expand the reasons.
Main concerns
- R: Several of my previous comments weren’t replied, being relevant for the definition of an expert system, and for the justification of this paper:
- Which type of inference engine does the expert system use? Which type of inference mechanism does it apply?
- Does de expert system proposed by the authors manage the uncertainty? If that’s the case, how?
- The authors must define which are the inputs and the outputs of the proposed expert system. Authors must also present a subset of rules.
- In Section 4.3, authors describe several simulation software, why? Is that necessary?
- After analysing both case studies, in what is the expert system helping? What is the output of the system? How is the expert system helping to design or to optimize the layout and reduce the time process?
- A discussion section must be added. What is the difference between this proposal and other? Why is this proposal better? What are the weaknesses of the system?
- Reviewer: What is the difference in the design of a shoe manufacturing plant and the design of another manufacturing plant? In the lines 28 – 29 it is said: ‘The position of the machines and the different processes steps can reduce the time of the whole process, and this is little studied in the literature’. This is a common and well-known problem when a manufacturing plant is designed. Why do the authors say that is little studied in the literature?
Author: Production plant optimization is a well-known area of knowledge in literature. It includes optimization of resources and layout design (distribution of elements along the factory). When we refer to this research as little studied, we are only talking about the shoe industry. This article applies known techniques to a non exploited industrial sector. It has been clarified on the text.
R: The authors didn’t justify this point enough in their manuscript, they just add new information in Lines 33-34. I’m not able to see the novelty of they proposal, as this is a well-known topic in the literature.
- R: In lines 36 – 37, the authors said: ‘For this reason, this research presents a more scientist method to design a manufacturing plant combined different elements such as expert systems and simulations. Therefore, it seems coherent to look for the solution in an expert system.’ What do the authors mean when they say a more scientist method? Why did the authors decide to use an expert system and no other approach?
A: Shoe factory design is usually made by experts that have been done it before. It is common to adjust some parameters using a trial and error technique over the real factory. This article presents a medium-size production plant where all elements are properly dimensioned (optimized). Here, a reader can find which machines will he need, how many workers should the factory have or how many pairs of shoes will be produced per day. The introduction has been slightly modified and Conclusions section.
R: The authors didn’t justify the use of an expert system. The appointed in Lines 40-47 is just general information which justify nothing about their choice. What information was added about this in the conclusions section?
- R: In lines 39 – 42, authors said: ‘These rules are mainly obtained from technical bibliography and expert knowledge. There are several techniques for acquiring knowledge for an expert system, such as those presented by A. Kidd [1].’ Which of the techniques for building the knowledge base of the Expert System, of those which are presented in the reference [1], did the authors use?
A: This text has been rewritten to clarify and specify some of the relevant features presented by Kidd and used in this research:
“There are several features such as flow description, relationship and dependencies be-tween processes, task times (including statistical distribution) or utilization rate that are required to build the Expert System. All of them are presented by A. Kidd in [1].”
R: How did the authors apply “flow description, relationship and dependencies between processes, task times (including statistical distribution) or utilization rate” for building their Expert System? Which statistical distribution were taken into account?
- R: In lines 47 – 49, authors said: ‘This work presents an expert system to support engineers in the conceptual and preliminary design phases of a shoe production factory, considering human and material resources and real processing times.’ Do the authors consider that an expert system is the same than a decision support system? I suggest the authors to review the paper of Pfeifer and Lüthi [1]. [1] Pfeifer, R.; Lüthi, H.-J. Decision Support Systems and Expert Systems: A Complementary Relationship? In Expert Systems and Artificial Intelligence in Decision Support Systems; Springer: Dordrecht, The Netherlands, 1987; pp. 41–51.
A: Thank you a lot for the recommendation, Pfeifer and Lüthi paper seems really interesting. We agree that this sentence is inaccurate and a new writing is proposed:
“This work presents an expert system, generated using the knowledge of an experienced group of people and where human and material resources and real processing times have been considered.”
R: This point must be rethought. The use of expert systems and decision support systems is complementary, as it is said in the paper by Pfeifer and Lüthi. It is obvious that this work presents a decision support system, which also uses expert system. Authors must understand the differences between both for the conceptualization of the system.
- R: Which design methodology are the authors following in the design process of the plant?
A: This paper presents an expert system following a Function Oriented Manufacturing Systems (FOMS). Product Oriented Manufacturing System (POMS) is an alternative to FOMS but after presenting both methodologies to our expert board, we decided to follow a FOMS.
R: Authors must clarify what Function Oriented Manufacturing Systems (FOMS) is in their manuscript, and why they decided to use this instead of other approach.
- R: In the Section 2 a literature review of some applications of expert systems in the industry is performed. Authors present some works without describing them in detail. Authors should add some lines about every work. Furthermore, some of the approaches presented do not use expert systems, so authors must point that, for example: ‘The work presented by [16] proposes a scheduling module focused on the short term in order to respond quickly to market needs and changes in a flexible manner. The module is composed of a finite capacity scheduler integrated with a new software based on the Analytical Hierarchy Process (AHP) decision support system. This module considers the aspects related to order importance as complexity or urgency and assign each order a priority.’ Please review the whole section. I consider also that the authors should make a table where they summarize all the works, one by one, as it will be easier for the reader. Authors should also add some lines where they describe the conclusions obtained after the review, explaining the general trends, when expert systems are used in the industry plant design.
A: The addition of a new table to list and summarize references used on the introduction is a good idea. However, we have slightly modified the text to explain references of Section 2 in more detail.
R: The authors didn’t pay attention in the revision of this point. They must differentiate the use of expert systems and other methods. Authors should also add some lines where they describe the conclusions obtained after the review, explaining the general trends, when expert systems are used in the industry plant design.
- R: In lines 90 – 93, it is said: ‘The last years, with the growth up of the artificial intelligence, it is combined with expert systems in order to improve the solutions in [7], [8] and [9] an artificial intelligence to improve some fields such as chemical process, food or agriculture is presented.’ Actually, expert systems are an artificial intelligence approach.
A: The text has been changed to be clearer and closer to the stated approach:
“The last years, with the growth up of the artificial intelligence, the expert systems are considering an approach of the artificial intelligence and they are use in order to improve their solutions in [17], [18] and [19].”
R: Ok. However English must be reviewed, there are some problems that must be addressed, I suppose that because of the use of an automatic translator (see Line 24).
- R: In lines 225-226, authors said: ‘The research is based on the combination between an expert system and a model simulation.’ Why is it based on that combination? What do the authors understand as a model simulation?
A: This paragraph has been rewritten to clarify that a simulation model is not used but a simulation process to validate the expert system proposed:
This research is based on an expert system design supported by discrete event simulation. Firstly, the expert knowledge collected is essential in order to create a data base. This data base will be used by the expert system to develop the design required. In this research a shoe manufacturing plant is designed using both tools. Secondly, a simulation of the production plant with it will be used to verify the solution. On the other hand, a high number of accurate information about the solution (task times, number of workers, etc.) will be gathered in the simulator.
R: How do you build the data base? How and where is this simulation performed? How is the solution verified?
- R: Authors must add a section where they present the technical grounds of the work and where they answer to several questions:
- What is it an expert system? What are the components of an expert system?
- Do the authors propose a decision support system? It that’s the case, what is a decision support system?
- The model presented in this manuscript needs to be considered within the information systems framework. Maybe the presented model and its implications could be understood as an artefact according to the design science research. In order to validate the contributions in the field of information systems, it is necessary that the authors evaluate their proposal according to the guidelines and principles proposed by Hevner et al. [2, 3]. [2] Hevner, A.R.; March, S.T.; Park, J.; Ram, S. Design science in information systems research. MIS Q. Manag. Inf. Syst. 2004, 28, 75–105, doi:10.2307/25148625. [3] Hevner, A.R.; Chatterjee, S. Design Research in Information Systems: Theory and Practice; Springer: New York, NY, USA, 2010; ISBN 978-1-4419-6107-5.
A: References provided are really interesting. Authors have overseen them to understand reviewer’s comment and some lines have been added to the text. References in the article to a decision support system has been removed to avoid misunderstandings. This research presents an expert system based on knowledge of experienced shoe factory managers. Simulations are used to validate the expert system and to show results of two use cases. These ideas have been introduced in several points of the manuscript.
About the components of the expert system, Figure 3 shows the diagram of elements and its connections.
R: This point must be revised. It is very important. Authors didn’t reply to my questions, that I’m asking again:
- What is it an expert system? What are the components of an expert system? If the Figure 3 presents the components, it must be explained in detail.
- Do the authors propose a decision support system? It that’s the case, what is a decision support system? I do not agree with the authors. I think that they are presenting a Decision Support System which is complemented with an expert system. (Again, see Pfeifer and Lüthi).
- The model presented in this manuscript needs to be considered within the information systems framework. Maybe the presented model and its implications could be understood as an artefact according to the design science research. In order to validate the contributions in the field of information systems, it is necessary that the authors evaluate their proposal according to the guidelines and principles proposed by Hevner et al. [2, 3]. [2] Hevner, A.R.; March, S.T.; Park, J.; Ram, S. Design science in information systems research. MIS Q. Manag. Inf. Syst. 2004, 28, 75–105, doi:10.2307/25148625. [3] Hevner, A.R.; Chatterjee, S. Design Research in Information Systems: Theory and Practice; Springer: New York, NY, USA, 2010; ISBN 978-1-4419-6107-5. This point is very important, authos must verify each of the guidelines.
- R: Sections 4.1 and 4.2 have the same title, Results and discussion. Why? Shouldn’t the method be presented before the results and discussion?
A: This mistake has been corrected.
R: Ok.
- R: Figure 3 must be explained in more detail, explaining each of the decisions made to build the diagram, and the information flow. How was the design of the model carried out?
A: Text has been added to better clarify the figure:
“Figure 3 represents the class diagram of the conceptual reference framework where the process area of the diagram has been described in six main process categories: Upper pieces cutting, Manufacturing, Quality test, Flash cutting, Cleaning/Polishing and Packaging. Such classification has been detailed following the formalization proposed by [30] and [31]. Each process categories are subclassified in the main subprocess describe in the follows section in the study. This subprocess are: sewing, leather buffing, pieces pre-shaping, cementing, lasting, milling, drilling and grinding. These sub-processes will be studied in depth in the research.”
R: Figure 3 needs to be explained in detail, a superficial description is not enough.
- R: What are the future lines of work?
A: A paragraph is added to present the future lines:
“The next step in the study process of the present work and as future lines, is the generalization of the proposed methodology to other types of manufacturing or production plants and to check the robustness of the methodology presented in other more complex environments.”
R: What do the authors understand for a more complex environment?

Author Response
Reviewer 2
The authors have performed a very superficial revision of their manuscript. They didn’t reply to all my comments (avoiding those which are more technical, and related with the definition of an expert system, or just replying to them without answering to what was pointed). The manuscript present serious flaws and problems, not only from the design perspective, but also from the conceptual one, being far away for being considered for publication. Moreover, it was really difficult to perform the review, as the authors did not provide the Lines of their changes on each of the comments.
For these reasons, this manuscript must be rejected again. I consider that this manuscript will not improve even with a Major Revision.
Below, it is possible to find some comments that expand the reasons.
Main concerns
- R: Several of my previous comments weren’t replied, being relevant for the definition of an expert system, and for the justification of this paper:
- Which type of inference engine does the expert system use? Which type of inference mechanism does it apply?
AThe implementation of the knowledge base system, the inference system and its relationships with the user and experts has been performed following the guideline of [39] (new reference added to the paper).
“The knowledge base system receives expert knowledge coded into facts, rules, heuristics and procedures. The expert system comprises three elements include database, knowledge base and inference engine. The general architecture of knowledge database and expert system is shown in Figure 2 of [39]. The database possesses information in terms of fact or heuristics based on user interest of specific problem domain. Knowledge base possesses domain knowledge expressed in terms of mathematical logic.”
The text presented above has been introduced on the paper to clarify this point (lines 269-274).
- Does de expert system proposed by the authors manage the uncertainty? If that’s the case, how?
A: The expert system proposed does not manage uncertainty. This is a very good idea that was evaluated during the research process, but it was dismissed due to its cost-benefit ratio in this project. The reference that as analyzed was: Jerzy W. Grzymala-Busse. Managing Uncertainty in Expert Systems. Verlag: Springer US, Print ISBN: 978-1-4613-6779-6, Electronic ISBN: 978-1-4615-3982-7
- The authors must define which are the inputs and the outputs of the proposed expert system. Authors must also present a subset of rules.
A: We have added the follow text to incorporate the information:
“In this research, the inputs which have been considering have been mainly the expert knowledge about the process, the dependencies between the different process and the mean time of the tasks. While the output is, above all, the design, and the best distribution of the process in the plant.”
- In Section 4.3, authors describe several simulation software, why? Is that necessary?
A: We present the different software that can be used for the development of the expert system and plant design and justify the use of one of them.
- After analysing both case studies, in what is the expert system helping? What is the output of the system? How is the expert system helping to design or to optimize the layout and reduce the time process?
A: The use of the expert system helps in the modelling of the manufacturing system, the dependencies between tasks in the process, the possible bottlenecks and the operators needed in each task.
- A discussion section must be added. What is the difference between this proposal and other? Why is this proposal better? What are the weaknesses of the system?
- Reviewer: What is the difference in the design of a shoe manufacturing plant and the design of another manufacturing plant? In the lines 28 – 29 it is said: ‘The position of the machines and the different processes steps can reduce the time of the whole process, and this is little studied in the literature’. This is a common and well-known problem when a manufacturing plant is designed. Why do the authors say that is little studied in the literature?
Author: Production plant optimization is a well-known area of knowledge in literature. It includes optimization of resources and layout design (distribution of elements along the factory). When we refer to this research as little studied, we are only talking about the shoe industry. This article applies known techniques to a non exploited industrial sector. It has been clarified on the text.
R: The authors didn’t justify this point enough in their manuscript, they just add new information in Lines 33-34. I’m not able to see the novelty of they proposal, as this is a well-known topic in the literature.
A: At present, the footwear industry has a very low level of automation, being practically manual. With the present study it is intended to make an approach to this automation, in this case of the design of the plant and the incorporation of different machines that automate the manufacturing process.
“For this main reason the shoe industry presents high difference with other industries, a reconversion of the industry is being carried out to automatized it, but this process is gradual, applying known techniques to a non-exploited industrial sector with the aim of full automatization.”
- R: In lines 36 – 37, the authors said: ‘For this reason, this research presents a more scientist method to design a manufacturing plant combined different elements such as expert systems and simulations. Therefore, it seems coherent to look for the solution in an expert system.’ What do the authors mean when they say a more scientist method? Why did the authors decide to use an expert system and no other approach?
A: Shoe factory design is usually made by experts that have been done it before. It is common to adjust some parameters using a trial and error technique over the real factory. This article presents a medium-size production plant where all elements are properly dimensioned (optimized). Here, a reader can find which machines will he need, how many workers should the factory have or how many pairs of shoes will be produced per day. The introduction has been slightly modified and Conclusions section.
R: The authors didn’t justify the use of an expert system. The appointed in Lines 40-47 is just general information which justify nothing about their choice. What information was added about this in the conclusions section?
A: We justified more the use of an expert systems with this paragraph:
“Due to the high degree of handicraft in this sector, the expert system is based on a knowledge base previously collected from expert workers in the footwear sector. The use of this knowledge is essential as there is no precedent in highly automated footwear manufacturing plants. This is the main reason for the use of the expert system in the development of this study.”
We make extensive use of the task times and utilisation rate to develop the plant layout and establish secure manufacturing flow and inter-task relationships and these features are included in the results and conclusions.
- R: In lines 39 – 42, authors said: ‘These rules are mainly obtained from technical bibliography and expert knowledge. There are several techniques for acquiring knowledge for an expert system, such as those presented by A. Kidd [1].’ Which of the techniques for building the knowledge base of the Expert System, of those which are presented in the reference [1], did the authors use?
A: This text has been rewritten to clarify and specify some of the relevant features presented by Kidd and used in this research:
“There are several features such as flow description, relationship and dependencies be-tween processes, task times (including statistical distribution) or utilization rate that are required to build the Expert System. All of them are presented by A. Kidd in [1].”
R: How did the authors apply “flow description, relationship and dependencies between processes, task times (including statistical distribution) or utilization rate” for building their Expert System? Which statistical distribution were taken into account?
A: The Flow description is used in the description of both cases in the section 5 and 6 as well as the dependencies between process. As explained in the text, the process is lineal and parallel process can not be made. The task time and the rate for building are used in the tables 1, 3 and 4. This information is essential to the expert system and to reach an optimal design. We use the mean in the times of the tasks, but the interval is very narrow. For this reason, it is not represented in the paper and only one data is used as the deviation is negligible.
- R: In lines 47 – 49, authors said: ‘This work presents an expert system to support engineers in the conceptual and preliminary design phases of a shoe production factory, considering human and material resources and real processing times.’ Do the authors consider that an expert system is the same than a decision support system? I suggest the authors to review the paper of Pfeifer and Lüthi [1]. [1] Pfeifer, R.; Lüthi, H.-J. Decision Support Systems and Expert Systems: A Complementary Relationship? In Expert Systems and Artificial Intelligence in Decision Support Systems; Springer: Dordrecht, The Netherlands, 1987; pp. 41–51.
A: Thank you a lot for the recommendation, Pfeifer and Lüthi paper seems really interesting. We agree that this sentence is inaccurate and a new writing is proposed:
“This work presents an expert system, generated using the knowledge of an experienced group of people and where human and material resources and real processing times have been considered.”
R: This point must be rethought. The use of expert systems and decision support systems is complementary, as it is said in the paper by Pfeifer and Lüthi. It is obvious that this work presents a decision support system, which also uses expert system. Authors must understand the differences between both for the conceptualization of the system.
A: Some text is added to improve the paragraph:
This work presents a decision support system which are combined with an expert system, using the knowledge of an experienced group of people to improve the final shoe plant design, supporting the design decisions and where human and material resources and real processing times have been considered.
R: Which design methodology are the authors following in the design process of the plant?
A: This paper presents an expert system following a Function Oriented Manufacturing Systems (FOMS). Product Oriented Manufacturing System (POMS) is an alternative to FOMS but after presenting both methodologies to our expert board, we decided to follow a FOMS.
R: Authors must clarify what Function Oriented Manufacturing Systems (FOMS) is in their manuscript, and why they decided to use this instead of other approach.
- R: In the Section 2 a literature review of some applications of expert systems in the industry is performed. Authors present some works without describing them in detail. Authors should add some lines about every work. Furthermore, some of the approaches presented do not use expert systems, so authors must point that, for example: ‘The work presented by [16] proposes a scheduling module focused on the short term in order to respond quickly to market needs and changes in a flexible manner. The module is composed of a finite capacity scheduler integrated with a new software based on the Analytical Hierarchy Process (AHP) decision support system. This module considers the aspects related to order importance as complexity or urgency and assign each order a priority.’ Please review the whole section. I consider also that the authors should make a table where they summarize all the works, one by one, as it will be easier for the reader. Authors should also add some lines where they describe the conclusions obtained after the review, explaining the general trends, when expert systems are used in the industry plant design.
A: The addition of a new table to list and summarize references used on the introduction is a good idea. However, we have slightly modified the text to explain references of Section 2 in more detail.
R: The authors didn’t pay attention in the revision of this point. They must differentiate the use of expert systems and other methods. Authors should also add some lines where they describe the conclusions obtained after the review, explaining the general trends, when expert systems are used in the industry plant design.
- R: In lines 90 – 93, it is said: ‘The last years, with the growth up of the artificial intelligence, it is combined with expert systems in order to improve the solutions in [7], [8] and [9] an artificial intelligence to improve some fields such as chemical process, food or agriculture is presented.’ Actually, expert systems are an artificial intelligence approach.
A: The text has been changed to be clearer and closer to the stated approach:
“The last years, with the growth up of the artificial intelligence, the expert systems are considering an approach of the artificial intelligence and they are use in order to improve their solutions in [17], [18] and [19].”
R: Ok. However English must be reviewed, there are some problems that must be addressed.
A: The text has been corrected
“In recent years, with the growth of artificial intelligence, expert systems are considered an approach to artificial intelligence and are used to improve their solutions in [17], [18] and [19].”
- R: In lines 225-226, authors said: ‘The research is based on the combination between an expert system and a model simulation.’ Why is it based on that combination? What do the authors understand as a model simulation?
A: This paragraph has been rewritten to clarify that a simulation model is not used but a simulation process to validate the expert system proposed:
This research is based on an expert system design supported by discrete event simulation. Firstly, the expert knowledge collected is essential in order to create a data base. This data base will be used by the expert system to develop the design required. In this research a shoe manufacturing plant is designed using both tools. Secondly, a simulation of the production plant with it will be used to verify the solution. On the other hand, a high number of accurate information about the solution (task times, number of workers, etc.) will be gathered in the simulator.
R: How do you build the data base? How and where is this simulation performed? How is the solution verified?
A: The building of the data base has been made with the knowledge of expert workers in the area as has been said previously in the paper. With this data the simulation is performed in order to obtain a better design, for last is verified by the company that require the study.
- R: Authors must add a section where they present the technical grounds of the work and where they answer to several questions:
- What is it an expert system? What are the components of an expert system?
- Do the authors propose a decision support system? It that’s the case, what is a decision support system?
- The model presented in this manuscript needs to be considered within the information systems framework. Maybe the presented model and its implications could be understood as an artefact according to the design science research. In order to validate the contributions in the field of information systems, it is necessary that the authors evaluate their proposal according to the guidelines and principles proposed by Hevner et al. [2, 3]. [2] Hevner, A.R.; March, S.T.; Park, J.; Ram, S. Design science in information systems research. MIS Q. Manag. Inf. Syst. 2004, 28, 75–105, doi:10.2307/25148625. [3] Hevner, A.R.; Chatterjee, S. Design Research in Information Systems: Theory and Practice; Springer: New York, NY, USA, 2010; ISBN 978-1-4419-6107-5.
A: References provided are really interesting. Authors have overseen them to understand reviewer’s comment and some lines have been added to the text. References in the article to a decision support system has been removed to avoid misunderstandings. This research presents an expert system based on knowledge of experienced shoe factory managers. Simulations are used to validate the expert system and to show results of two use cases. These ideas have been introduced in several points of the manuscript.
About the components of the expert system, Figure 3 shows the diagram of elements and its connections.
R: This point must be revised. It is very important. Authors didn’t reply to my questions, that I’m asking again:
- What is it an expert system? What are the components of an expert system? If the Figure 3 presents the components, it must be explained in detail.
A: Figure 3 represents de main components of the expert system and the main branches and process are described. All other processes are not part of the expert system in this study.
- Do the authors propose a decision support system? It that’s the case, what is a decision support system? I do not agree with the authors. I think that they are presenting a Decision Support System which is complemented with an expert system. (Again, see Pfeifer and Lüthi).
A: The research is supported by a Decision Support System combined with an expert system. This is added during the study.
The model presented in this manuscript needs to be considered within the information systems framework. Maybe the presented model and its implications could be understood as an artefact according to the design science research. In order to validate the contributions in the field of information systems, it is necessary that the authors evaluate their proposal according to the guidelines and principles proposed by Hevner et al. [2, 3]. [2] Hevner, A.R.; March, S.T.; Park, J.; Ram, S. Design science in information systems research. MIS Q. Manag. Inf. Syst. 2004, 28, 75–105, doi:10.2307/25148625. [3] Hevner, A.R.; Chatterjee, S. Design Research in Information Systems: Theory and Practice; Springer: New York, NY, USA, 2010; ISBN 978-1-4419-6107-5. This point is very important, authos must verify each of the guidelines.
A: The knowledge of design science research (DSR) can have applications for improving expert systems (ES) development research. Although significant progress of utilising DSR has been observed in particular information systems design – such as decision support systems (DSS) studies – only rare attempts can be found in the ES design literature. It has been demonstrated DSR, in terms of both problem class and functions of ES artefacts, can help ES designers and researchers to address issues for designing information system solutions [A].
The idea pointed by the reviewer is interesting and uncommon. It is indeed a not very much explored/exploited approach, but it is a different way to tackle the problem. This research is focused in a different manner, not better, not worse.
[A] Miah, Shah & Genemo, Hussein. (2016). A Design Science Research Methodology for Expert Systems Development. Australasian Journal of Information Systems. 20. 10.3127/ajis.v20i0.1329.
- R: Sections 4.1 and 4.2 have the same title, Results and discussion. Why? Shouldn’t the method be presented before the results and discussion?
A: This mistake has been corrected.
R: Ok.
- R: Figure 3 must be explained in more detail, explaining each of the decisions made to build the diagram, and the information flow. How was the design of the model carried out?
A: Text has been added to better clarify the figure:
“Figure 3 represents the class diagram of the conceptual reference framework where the process area of the diagram has been described in six main process categories: Upper pieces cutting, Manufacturing, Quality test, Flash cutting, Cleaning/Polishing and Packaging. Such classification has been detailed following the formalization proposed by [30] and [31]. Each process categories are subclassified in the main subprocess describe in the follows section in the study. This subprocess are: sewing, leather buffing, pieces pre-shaping, cementing, lasting, milling, drilling and grinding. These sub-processes will be studied in depth in the research.”
R: Figure 3 needs to be explained in detail, a superficial description is not enough.
A: Figure 3 is described the process which are study for the plant design. We have added a text to clarify:
“This study is focused on in the manufacturing process. The other processes as they are not part of the manufacturing process, will not be part of the design in the current study.”
- R: What are the future lines of work?
A: A paragraph is added to present the future lines:
“The next step in the study process of the present work and as future lines, is the generalization of the proposed methodology to other types of manufacturing or production plants and to check the robustness of the methodology presented in other more complex environments.”
R: What do the authors understand for a more complex environment?
A: In this specific case, will be a full automatized manufacturing system. We have added a new text to clarify:
“When the full process is automatized with robotics systems presenting news dependencies.”
Reviewer 3 Report
Changes are made and the overall manuscript is improved
Author Response
Thank you very much for your comments, changes have been made to improve the paper.
Reviewer 4 Report
All my comments have been well responded.
Author Response

(The authors gave the same response as above.)

Round 3
Reviewer 2 Report
The problems pointed in my last two reviews weren’t addressed. Most of my comments were neglected or replied in an inappropriate way, ignoring those which are fundamental. The proposed expert system was not clarified, it is not possible to understand what is aimed to. I still do not know what are the inputs of the expert system, the knowledge base (what rules they are using), the inference engine used or its outputs…Moreover, I’ve asked the authors twice to add a discussion section, which was not added.
The authors didn’t improve the paper after the two last review rounds. I suggest again to reject it.